# Exploring the Genetic Relationship Between Type 2 Diabetes and Cardiovascular Disease: A Large-Scale Genetic Association and Polygenic Risk Score Study

**DOI:** 10.3390/biom14111467

**Published:** 2024-11-18

**Authors:** Ziwei Yao, Xiaomai Zhang, Liufei Deng, Jiayu Zhang, Yalu Wen, Deqiang Zheng, Long Liu

**Affiliations:** 1Academy of Medical Sciences, Shanxi Medical University, No 56 Xinjian South Road, Yingze District, Taiyuan 030001, China; yzw000302@gmail.com (Z.Y.); libertas.wheat@gmail.com (X.Z.); 2Department of Health Statistics, School of Public Health, Shanxi Medical University, No 56 Xinjian South Road, Yingze District, Taiyuan 030001, China; dengwenhui1212@gmail.com (L.D.); zjiayu0217@gmail.com (J.Z.); 3Department of Statistics, University of Auckland, 38 Princes Street, Auckland Central, Auckland 1010, New Zealand; y.wen@auckland.ac.nz; 4Department of Epidemiology and Health Statistics, School of Public Health, Capital Medical University, Beijing 100054, China

**Keywords:** type 2 diabetes, cardiovascular disease, polygenic risk scores, genetic correlation, phenotypic association analyses

## Abstract

Type 2 diabetes (T2D) is often comorbid with cardiovascular diseases (CVDs). The direction of causation between T2D and CVD is difficult to determine; however, there may be a common underlying pathway attributable to shared genetic factors. We aimed to determine whether there is a shared genetic susceptibility to T2D and CVD. This study utilizes large-scale datasets from the UK Biobank (UKB) and DIAGRAM consortium to investigate the genetic association between T2D and CVD through phenotypic association analyses, linkage disequilibrium score (LDSC) analysis, and polygenic risk score (PRS) analysis. LDSC analysis demonstrates significant genetic associations between T2D and various CVD subtypes, including angina, heart failure (HF), myocardial infarction (MI), peripheral vascular disease (PVD), and stroke. Although the genetic association between T2D and atrial fibrillation (AF) was not significant, individuals in the high-T2D PRS group had a significantly increased risk of CVD. These findings suggest a common genetic basis and suggest that genetic susceptibility to T2D may be a potential predictor of CVD risk.

## 1. Introduction

Cardiovascular diseases (CVDs) are a broad term encompassing a range of related conditions, typically including atrial fibrillation (AF) [1], angina [2], heart failure (HF) [3], myocardial infarction (MI) [4], peripheral vascular disease (PVD) [5], and stroke [6]. CVD ranks among the leading causes of death globally and poses a significant public health challenge [7]. These conditions not only diminish patients’ quality of life but also impose a substantial burden on healthcare systems. According to the Global Burden of Disease report, CVD is a major contributor to mortality and permanent disability, particularly affecting older populations [8]. However, despite the known differences in CVD risk between males and females, most studies examining the impact of individual risk factors have traditionally focused on male subjects [9]. As the global population ages and lifestyle factors evolve, the incidence and prevalence of CVD have steadily increased [10]. Unhealthy lifestyle choices, such as smoking, insufficient physical activity, and unhealthy dietary habits, exacerbate the risk of developing CVD [11,12]. Despite recent advancements in managing cardiovascular risk factors, type 2 diabetes (T2D) remains a notable complication associated with CVD [13]. Individuals with T2D exhibit a 1.5 to 2-fold increased risk of CVD compared to those without the condition [14]. Epidemiological studies indicate that T2D and CVD share several clinical risk factors, and hyperglycemia itself significantly elevates the risk of both cardiovascular disease and related mortality [15]. Investigating the relationship between T2D and CVD is a crucial strategy for mitigating cardiovascular risk. Research on this association will play an essential role in the development of preventive and therapeutic measures for CVD. By predicting the risk of T2D in patients with CVD, we can implement preventive strategies, reduce healthcare costs, and enhance the quality of care.

Recent research has demonstrated that the incidence of T2D is influenced not only by environmental factors but also significantly by genetic factors [16,17]. Furthermore, there exists a strong association between these genetic factors and the risk of CVD [18]. The advancement of large-scale genome-wide association studies (GWASs) has enhanced our understanding of the genetic variations linked to both T2D and CVD [19,20]. Utilizing GWAS data, we can estimate the genetic correlation between these two traits through linkage disequilibrium score regression (LDSC) analysis [21]. The LDSC method effectively quantifies the contribution of genetic variation to traits and assesses the genetic correlation between different traits, thereby revealing their shared genetic basis.

Researchers have developed the polygenic risk score (PRS) method [22,23,24] to quantify an individual’s genetic risk, providing a powerful tool for predicting the likelihood of T2D and CVD. One of the most commonly used methods for constructing PRSs is the clumping and thresholding (C + T) approach, also known as pruning and thresholding (P + T) This method involves two filtering steps. The first step involves the clumping of single nucleotide polymorphisms (SNPs) using a linkage disequilibrium (LD)-driven clumping procedure [25], which retains only those SNPs that exhibit weak correlations. Each clump contains all SNPs located within 250 kb of the index SNPs, with the degree of LD determined by a specified pairwise correlation (r^2^). Subsequently, SNPs with p-values obtained from a disease-related GWAS that exceed a predetermined threshold are excluded. The C + T method is recognized as one of the simplest and most intuitive techniques for constructing PRS. Two common software programs, PLINK and PRSice, can be employed to implement the C + T method. Recently, Choi et al. developed a new software, PRSice-2, which is available at https://www.precis-2.org/ [26], accessed on 10 July 2024. This method has demonstrated greater computational efficiency and scalability compared to other PRS software while maintaining comparable predictive power.

To investigate the genetic relationship between T2D and CVDs, our study first confirmed the association between these conditions through phenotypic association analyses using data from the UK Biobank (UKB). We then assessed the existence of a shared genetic relationship by implementing LDSC to estimate genetic correlations between the traits. Finally, we employed results from GWASs to analyze genetic associations across traits using a PRS approach. The development and progression of both T2D and CVD are influenced by a complex interplay of genetic susceptibility and lifestyle factors impacting metabolic health [27]. However, the genetic susceptibilities related to T2D and the lifestyle factors that contribute to CVD risk remain unclear, and the interactions between the two and metabolic health are also unknown. Despite significant advancements in the diagnosis and treatment of CVD in recent years, the intricate pathological mechanisms and diverse manifestations of the disease continue to present challenges for clinical management. Understanding the genetic susceptibility to T2D and its association with CVD is crucial for the development of more effective early prevention strategies and personalized medicine. Future research will further explore these interrelated factors to improve the management and treatment of T2D and CVD. Comprehensive research into the causes, prevention, and treatment strategies for CVD is essential to reduce global mortality and improve public health.

## 2. Materials and Methods

### 2.1. Data Sources

The primary data for this study were sourced from the UKB database, a comprehensive biomedical resource [28]. As part of a prospective epidemiological research initiative, the UKB recruited approximately 500,000 healthy volunteers aged 40 to 69 from across the United Kingdom between 2006 and 2010. The database encompasses extensive participant information, including demographic data, lifestyle factors, medical history, and family history. Furthermore, physical measurements, imaging assessments, and biochemical tests were conducted. All participants provided informed consent prior to data collection. The UKB cohort study received ethical approval from the North West Multi-center Research Ethics Committee (MREC) [29]. This research was conducted using resources from the UKB under approved application number 95259.

The T2D data utilized in this study were derived from the DIAGRAM (DIAbetes Genetics Replication And Meta-analysis) consortium [30]. This consortium is committed to elucidating the genetic underpinnings of T2D through extensive studies, primarily concentrating on samples of European ancestry. The T2D summary statistics provided by the DIAGRAM consortium constitute one of the largest publicly accessible GWAS datasets, amalgamating data from 32 prior GWASs involving European populations, which include 74,124 cases of T2D and 824,006 non-T2D controls. The summary statistics for AF, angina, HF, MI, PVD, and stroke were sourced from the GWAS catalog [31]. All GWAS summary statistics were derived from European populations, with specific study designs and data characteristics described in Table 1.

### 2.2. Phenotypic Association Analyses in UKB

In this study, we utilized baseline data from participants’ initial assessments conducted between 2006 and 2010 within the UKB. The study population was limited to individuals who satisfied the UKB’s internal genetic quality control criteria (UKB Field 22020) and self-identified as White British (UKB Field 21000). A total of 365,008 eligible individuals were included in the analysis to assess the phenotypic associations between T2D and CVD-related conditions. T2D, AF, angina, HF, MI, PVD, and stroke were identified through diagnoses recorded in the UKB’s hospital inpatient data (using ICD codes) and through self-reports (see Appendix A).

Logistic regression models were employed to assess the phenotypic associations between T2D and various conditions, including AF, angina, HF, MI, PVD, and stroke. This analysis was conducted solely among the White British participants (*N* = 365,008) from the UKB to ensure the consistency and reliability of the results.

### 2.3. Covariates

Relevant variables were extracted based on demographic characteristics, hospital admission examinations, lifestyle factors, and biological variables. The demographic characteristics included age, sex, and educational attainment. The years of education were inferred from these variables using the International Standard Classification of Education (ISCED-97) applied to the United Kingdom’s educational qualifications. Specifically, a “College or University degree” was coded as 20 years of education, “A levels/AS levels or equivalent” as 13 years, “O levels/GCSEs or equivalent” as 10 years, “CSEs or equivalent” as 10 years, “NVQ or HND or HNC or equivalent” as 19 years, “Other professional qualifications, e.g., nursing, teaching” as 15 years, and “None of the above” as 7 years [32]. Hospital admission examinations encompassed the assessment of body mass index (BMI), as well as systolic and diastolic blood pressure. Lifestyle factors evaluated included smoking status and alcohol consumption. The biological variables analyzed comprised cholesterol levels, high-density lipoprotein (HDL) cholesterol, low-density lipoprotein (LDL) cholesterol, triglycerides (TG), C-reactive protein (CRP), glycated haemoglobin (HbA1c), Apolipoprotein A (Apo-A), Apolipoprotein B (Apo-B), red blood cell count (RBC), white blood cell count (WBC), platelet count (PLT), creatinine, and vitamin D. These biomarkers are particularly valuable for the assessment and monitoring of various health conditions, notably CVDs, diabetes, and other metabolic disorders. We characterized the baseline characteristics of the participants using mean ± standard deviation (SD) for normally distributed continuous variables, median with interquartile range (IQR) for non-normally distributed variables, and counts (percentages) for categorical variables.

### 2.4. LD Score Genetic Correlation Analysis

The genetic correlations between two traits were estimated using LDSC [33]. The genetic correlation (r_g_) quantifies the association between the genetic effects of two traits while controlling for environmental influences. In the LD score regression analysis, we utilized LD scores derived from the 1000 Genomes European dataset (accessible at https://data.broadinstitute.org/alkesgroup/LDSCORE/eur_w_ld_chr.tar.bz2, accessed on 20 July 2024) as a reference panel to appropriately weight the regression coefficients for the correlated SNPs. LDSC is particularly appropriate for large-scale studies as it does not require individual-level data, making it more efficient than methods such as Genome-wide Complex Trait Analysis (GCTA) and Local Analysis of [co]Variant Association (LAVA), both of which rely on individual-level data to estimate genetic variation and trait correlations. While GCTA focuses on estimating these genetic parameters, LAVA specifically analyzes the local ancestry of admixed populations to investigate the influence of genetic variation on ancestral origins at specific loci [34,35]. In contrast, LDSC is better suited to elucidate the shared genetic architecture between T2D and CVD based on GWAS summary statistics.

### 2.5. Genotyping and Quality Control

We utilized the UKB Axiom array in PLINK format and converted it to VCF format using plink2 (v.2.00a3.1LM), resulting in 784,256 variant sites across autosomes for 488,377 individuals. This subset of single nucleotide polymorphisms (SNPs) was employed for the analysis. Initially, SNPs were excluded based on the following criteria: (1) individuals with missing rate greater than 0.1; and (2) SNPs with missing rate greater than 0.1. For the retained SNPs after this quality control step, missing SNPs were imputed using the plink2R package in R (version 4.2.0). Further quality control was conducted on the imputed data, excluding SNPs with a missing rate greater than 0.01, Hardy–Weinberg equilibrium p-value less than 1 × 10^−6^, minor allele frequency (MAF) less than 0.05, and individuals with missing rate greater than 0.01. Following these two quality control steps, 301,096 SNPs were retained in the UKB dataset.

### 2.6. Polygenic Risk Scores

PRSs quantify an individual’s genetic predisposition to a specific trait by summing the products of genotype dosage for each variant and its corresponding effect size. We selected PRS for risk prediction because of its ability to efficiently quantify individual genetic risk in large populations without being limited to individual gene associations. PRS is better suited to provide a comprehensive risk assessment than localized analysis methods, thus providing a more practical predictive tool for understanding complex diseases such as T2D and CVD. In this study, we generated the PRS for T2D using samples from the UKB and the PRSice software, leveraging GWAS summary statistics from the DIAGRAM consortium for T2D, while excluding UKB samples. The PRS was constructed across a range of *p*-value thresholds from 5 × 10^−8^ to 0.05, applying a linkage disequilibrium (LD) pruning parameter of 0.2 within a 250 kb window. For 365,008 participants, the PRS was calculated by multiplying the genotype dosage by its corresponding weight and summing these values across all variants. We standardized the PRS and selected the version that explained the best phenotypic variance (R^2^).

### 2.7. Statistical Analysis

Figure 1 provides an overview of the research. The PRS with the best variance explained (R^2^) in T2D was divided into quartiles according to its distribution. We computed odds ratios (ORs) for each quartile using logistic regression models with CVD as the outcome variable. The first quartile, representing the lowest genetic risk for T2D, served as the reference group, and ORs were calculated by comparing this group to the other three quartiles. The logistic regression models were adjusted sequentially with different sets of covariates, resulting in four distinct models.

In the logistic regression model, CVD was designated as the outcome variable, and covariates were adjusted sequentially. We calculated the odds ratio (OR) between the first quartile (i.e., the group with the lowest genetic risk of T2D) and the other three quartiles. To assess the association between PRS and CVD risk, we constructed five stepwise adjusted logistic regression models, with PRS analyzed as the primary independent variable. Model 1 included only PRS variables to assess the unadjusted associations between different PRS quartiles (Q1–Q4) and CVD risk, and served as a baseline model to observe the effect of PRS on CVD risk. Model 2 builds on Model 1 by adding demographic characteristics as covariates, including age, sex, and education level. This adjustment took into account potential demographic factors associated with CVD risk to minimize their potential confounding effects on PRS outcomes. Model 3 further includes hospital admission examination data, such as BMI and blood pressure, to Model 2 in order to control for the impact of prior health conditions on CVD risk. Extending from Model 3, Model 4 incorporates adjustments for lifestyle factors, including behavioral variables such as smoking status and alcohol consumption, to evaluate the independent effect of PRS after accounting for lifestyle influences. Finally, Model 5 introduces adjustments for biological variables, allowing us to determine whether PRS remains significantly associated with CVD risk after controlling for key biological indicators. Additionally, we conducted corresponding analyses stratified by sex. We also replicated these analyses for AF, angina, HF, MI, PVD, and stroke, with the results presented in the Appendix A.

## 3. Results

A total of 369,008 individuals were included in the study, among whom 12,977 were diagnosed with T2D and 32,963 with CVD. A summary of the baseline characteristics of the study population is presented in Table 2. The T2D and CVD case groups exhibited significant differences in various demographic and clinical characteristics, including age, sex ratio, education level, lifestyle factors, hospitalization examination results, and biomarker levels (Appendix A). Notably, the case group demonstrated a higher average age, a greater proportion of males, and a lower education level. Additionally, there was a slightly higher prevalence of smokers within this group. The case group also presented elevated cardiovascular risk indicators, such as BMI, blood pressure, and red and white blood cell counts, alongside lower levels of HDL cholesterol and vitamin D. These differences provide an important foundation for association studies exploring the relationship between T2D and CVD.

We investigated the phenotypic associations between T2D and various CVDs, including AF, angina, HF, MI, PVD, and stroke, as well as the phenotypic association between T2D and overall CVD. The findings are depicted in Figure 2. The analysis indicated a significant association between T2D and CVDs, with an OR of 2.60 (95% confidence interval [CI] 2.18–2.72). Statistically significant associations were also observed between T2D and angina, HF, MI, and stroke (p < 0.001). We found odds ratios of 0.88 (95% CI: 0.70–1.10) for atrial fibrillation (AF) and 0.63 (95% CI: 0.37–0.99) for peripheral vascular disease (PVD) among individuals with T2D, suggesting a potential negative correlation or protective effect of T2D concerning AF and PVD.

LDSC analysis identified significant genetic correlations between T2D and various cardiovascular diseases (CVDs). Notably, there were positive genetic correlations between T2D and angina (r_g_ = 0.38, SE = 0.05, p < 0.001), HF (r_g_ = 0.42, SE = 0.15, p < 0.001), MI (r_g_ = 0.33, SE = 0.04, p < 0.001), PVD (r_g_ = 0.3212, SE = 0.11, p < 0.001), and stroke (r_g_ = 0.43, SE = 0.08, p < 0.001) (see Appendix A). These findings imply a potential genetic overlap and shared biological pathways among these conditions. In contrast, the genetic correlation between T2D and AF was weak and not statistically significant (r_g_ = 0.03, SE = 0.04, p = 0.44) (see Appendix A). Overall, these results suggest that, with the exception of AF, T2D exhibits significant genetic associations with other CVDs.

In this study, we utilized the PRS for T2D to predict the risk of CVDs. We evaluated the association between different PRS quartiles and CVD risk using a series of logistic regression models, incorporating progressively adjusted confounding factors. Additionally, we stratified the analysis by sex to explore potential differences in risk between males and females (Figure 3).

Model 1 included only the PRS variable, revealing a significant positive correlation between PRS quartiles and CVD risk. Compared to the lowest quartile (Q1), individuals in the highest quartile (Q4) exhibited a significantly elevated risk of developing CVD (OR = 1.33, 95% CI: 1.29–1.38, p < 0.001; Table 3). Stratified by sex, the odds ratio for Q4 was 1.34 (95% CI: 1.29–1.40) in males and 1.32 (95% CI: 1.25–1.40) in females, indicating a similar increased risk in both sexes (Figure 3). As PRS increased from Q1 to Q4, the risk of CVD progressively escalated in both males and females, demonstrating a clear predictive effect of PRS on CVD risk across genders. In Models 2 through 4, we adjusted for demographic characteristics, hospital admission examinations, and lifestyle factors. Although the effect of PRS slightly diminished following these adjustments, it remained statistically significant. For example, in Model 4, the odds ratio (OR) for PRS Q4 was 1.27 (95% CI: 1.23–1.31, p < 0.001; Table 3), indicating that a higher PRS is still associated with an increased risk of CVD. Stratified by sex, the OR for Q4 in Model 4 was 1.28 (95% CI: 1.23–1.33) in males and 1.25 (95% CI: 1.19–1.33) in females (Figure 3), further confirming that a higher PRS is consistently associated with an increased risk of CVD in both sexes. Finally, in Model 5, after adjusting for biological variables, the association between PRS and CVD risk was further attenuated but continued to be statistically significant. The OR for PRS Q4 was 1.07 (95% CI: 1.03–1.11, p < 0.001; Table 3). Stratified analysis revealed ORs of 1.06 (95% CI: 1.01–1.10) in males and 1.09 (95% CI: 1.03–1.16) in females (Figure 3), suggesting that even after controlling for a comprehensive range of biological markers, PRS remains a crucial predictor of CVD risk across both genders.

## 4. Discussion

In this study, we utilized large-scale genetic data to conduct an in-depth exploration of the genetic relationships between T2D and CVD. Understanding this relationship is crucial, as it can provide insights into common pathophysiological mechanisms and inform potential therapeutic strategies. First, we performed the phenotypic association analyses and found a significant association between T2D and CVD, which establishes a solid foundation for further genetic research. To validate this finding and explore the underlying genetic mechanisms, we employed two different genetic analysis methods: LDSC analysis and PRS analysis. LDSC analysis offers a broader perspective on genetic correlations at the genome-wide level, further supporting the genetic link between T2D and CVD. The PRS method assesses the relationship between genetic variation and disease risk by calculating an individual’s genetic risk score. Our findings indicate that genetic risk scores for T2D play a significant role in predicting CVD risk, suggesting that these diseases may share a portion of their genetic basis. Additionally, through this triangulation method, we can achieve a more comprehensive understanding of the role of genetic factors in the relationship between T2D and CVD.

Through an analysis of the extensive UKB dataset, we validated significant phenotypic associations between T2D and various CVDs, including angina, HF, MI, and stroke. This finding aligns with the study conducted by Aune, Georgakis et al. [36,37,38,39]. Our results suggest that while there is a significant association between T2D and several CVDs, the positive correlation between T2D and AF is relatively weak. Some studies indicate that the genetic association between T2D and AF may not be significant, possibly due to the mediating role of systolic blood pressure in this relationship [40]. Although T2D patients have a significantly increased risk of both macrovascular and microvascular complications in the development of AF, there is no direct evidence linking T2D to an increased prevalence of AF [41]. Nevertheless, some studies have shown that T2D may significantly increase the risk of mortality, coronary heart disease, and HF in patients with AF. This risk is particularly pronounced in elderly patients with T2D, where AF significantly raises the absolute risk of cerebrovascular events [42]. Thus, while T2D may not directly induce AF, effective blood glucose management remains crucial for AF patients to mitigate the risk of complications. Similarly, existing evidence does not support the notion that T2D directly causes PVD. However, research indicates that patients with both T2D and PVD experience a significantly greater risk of all-cause mortality and cardiovascular death compared to those with CVD alone [43]. Thus, although T2D does not seem to play a significant direct role in the onset of PVD, clinical management should still prioritize monitoring blood glucose levels in T2D patients to reduce the risk of comorbidities.

In recent years, extensive research has identified common risk factors for CVD in individuals with T2D [44,45,46,47]. Additionally, several epidemiological studies have explored the relationship between T2D and CVD, providing evidence of an association [48]. However, genetic association analyses between T2D and CVD remain limited. To our knowledge, this is the first comprehensive assessment of the genetic associations between T2D and CVD. In our study, we estimated genetic correlations using LDSC analysis on large-scale data from the UKB and the DIAGRAM consortium. Our findings indicate a significant genetic association between T2D and various CVDs, including AF, HF, MI, PVD, and stroke. This correlation further supports the potential existence of a shared genetic basis between T2D and CVD. Our study found that the strong genetic correlation between T2D and AF, MI, and stroke suggests a closer genetic relationship among these conditions. This finding is consistent with recent trends in the study of complex disease comorbidities, suggesting that the genetic basis of different chronic diseases may have important implications for public health and clinical practice [49,50,51]. Our genetic correlation analyses are based on GWAS summary data and are often consistent with results from individual phenotypes, such as genetic overlap between T2D and AF, HF, MI, and stroke. By using large-scale datasets, we confirmed a potential genetic association between T2D and CVDs.

In our analysis of PRS, we found that a high-T2D PRS was significantly associated with an increased risk of CVD. Although this association was attenuated after adjusting for demographic characteristics, hospital admission examinations, lifestyle factors, and biological variables, it remained statistically significant. This finding suggests that genetic susceptibility to T2D not only predicts the occurrence of T2D itself but is also closely associated with the risk of CVD. The results of our PRS analysis indicate that genetic susceptibility to T2D could serve as a potential tool for predicting CVD risk. In the future, personalized risk assessments and prevention strategies may be implemented to reduce the incidence of CVD among T2D patients. Additionally, the gender differences observed in our study warrant further attention. We found that male PRSs may predict CVD more effectively than gender-independent PRSs. There are well-known differences in both the incidence and presentation of CVD between the sexes, suggesting that sex-specific PRS may be better than sex-agnostic PRS. However, most GWASs are limited to autosomal chromosomes, and thus far, no associations with CVD have been detected on the X chromosome [52]. Nevertheless, there may be sex-specific effects at the same autosomal loci; studies have demonstrated that the sex-specific PRSs have superior predictive power compared to existing PRS that do not account for these effects. From a public health perspective, this study highlights the genetic association between T2D and CVD, suggesting that the comorbidity of these diseases should be considered in the development of comprehensive prevention strategies for T2D and CVD. These strategies may include more rigorous surveillance of high-risk groups and personalized interventions for individuals at elevated genetic risk.

The findings of this study have significant implications for the clinical management of individuals with T2D. Considering the increased risk of CVDs in T2D patients, particularly related to angina, HF, MI, PVD, and stroke, early identification and intervention for these risks within clinical practice may help reduce the burden of CVDs among T2D patients. In addition, results from the PRS analysis indicate that genetic susceptibility to T2D could serve as a valuable tool for predicting CVD risk. Future initiatives should prioritize personalized risk assessment and preventive strategies aimed at reducing the incidence of CVD in T2D populations. From a public health perspective, this study underscores the complex genetic relationship between T2D and CVDs, suggesting that comprehensive prevention strategies for both T2D and CVDs should take their comorbidity into account. Such strategies may encompass more intensive monitoring for high-risk groups and tailored interventions for individuals exhibiting elevated genetic risk.

However, there are several limitations of this study that warrant further discussion. First, some of the self-reported data utilized may be biased, particularly in relation to lifestyle factors such as smoking and alcohol consumption. Self-reported data are affected by factors such as respondents’ memory bias and social desirability bias, and may not accurately reflect actual behavior. To enhance data quality, future studies should prioritize the use of more reliable and clinically validated data sources, such as medical records or biospecimens, to minimize potential biases and improve the credibility of the findings. Second, although this study adjusted for lifestyle factors, mainly including smoking status and alcohol consumption, there are other potential environmental or lifestyle factors (e.g., diet and physical activity specifics) that may play an important role in the association between CVD and T2D. Future studies should collect these additional lifestyle data whenever possible and incorporate them into their analyses. Furthermore, as the study population is mainly European, the findings may not generalize to other populations. Some studies have identified genetic differences between populations in different regions, which may influence the associations between diseases [53,54]. Future research should aim to replicate this study in ethnically and geographically diverse populations to enhance the validity and applicability of the results globally. Finally, this study has notable limitations regarding causation. As a correlational study, this research revealed a genetic association between T2D and CVD; however, it does not establish a direct causal relationship. While a strong association between genetic susceptibility and CVD risk was identified, it remains uncertain whether T2D directly elevates CVD risk or if these diseases share common genetic or environmental factors. Future prospective studies should utilize causal inference models to investigate the causation between T2D and CVD more thoroughly and to help identify direct links or underlying mechanisms between these two conditions.

With advancements in sequencing technologies, subsequent studies could utilize whole genome sequencing (WGS) to delve deeper into the mechanisms underlying the association between T2D and CVD, thereby identifying potential new genetic variants and biomarkers. Exploring the complex association between T2D and AF is another crucial direction for future research. Although this study did not reveal significant genetic correlations, it is essential to validate these findings in large-scale datasets, and in particular, comprehensive data on environmental factors and lifestyle should be included to better elucidate the impact of these factors on the association between T2D and AF. Although PRS models have demonstrated enhanced potential in predicting CVD risk, their clinical application warrants further investigation. Future research should focus on developing targeted strategies for the effective integration of PRS scores into clinical practice, as well as exploring methods to combine these scores with traditional clinical risk factors to achieve a more comprehensive risk assessment. Given the complex interplay between genetic and environmental factors in T2D and CVD, future research could further improve the understanding of these relationships through statistical modeling of gene–environment interactions. Such models can help explore the moderating effects of different environmental factors on PRS and provide more basis for personalized risk assessment and intervention strategies. Furthermore, as PRS models are developed and validated based on a single cohort, there is a risk of overfitting, particularly concerning the UK Biobank sample. To enhance the reliability of the PRS model, it is imperative that future studies validate these findings across different datasets or external cohorts. This will help to validate the ability of the PRS model to generalize across different populations and provide stronger evidence to support its wider application.

## 5. Conclusions

In conclusion, this study, which utilizes large-scale population data, confirms the significant association between T2D and CVDs. It is the first to investigate the genetic basis of this association using LDSC analysis in a substantial sample. These findings not only enhance our understanding of the comorbidity progression of CVD in patients with T2D but also offer valuable insights for the development of more effective intervention strategies in both clinical and public health practice. However, this study does have certain limitations, and future research is needed to further validate these findings and to explore how genetic information can be utilized for disease risk management and personalized healthcare.

## Figures and Tables

**Figure 1 biomolecules-14-01467-f001:**
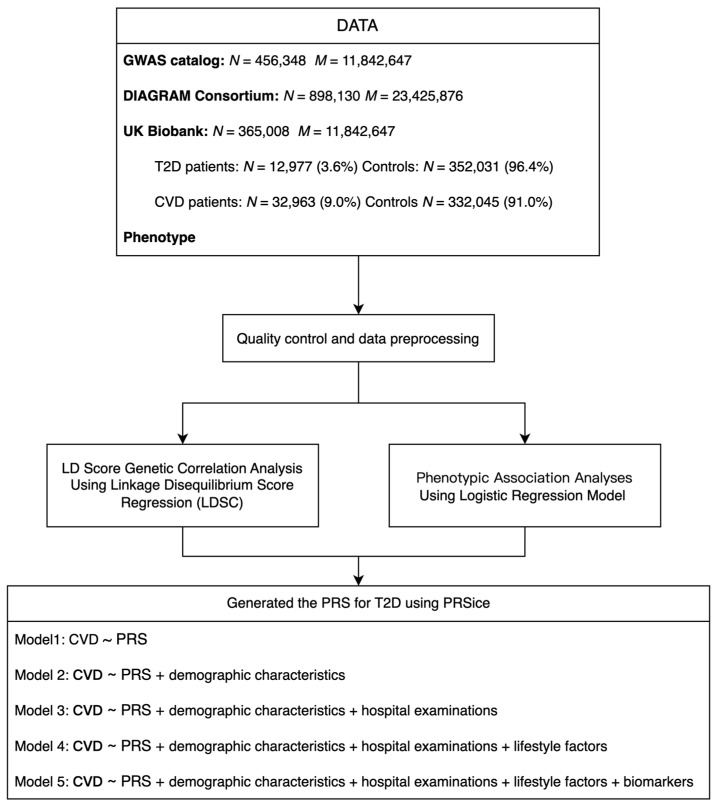
The flow chart of this study.

**Figure 2 biomolecules-14-01467-f002:**
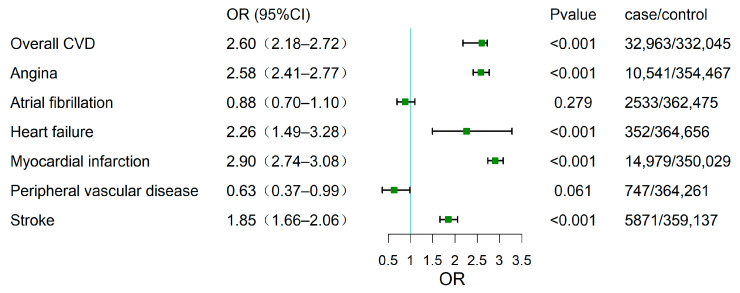
Forest plot illustrating the odds ratios (OR) and 95% confidence intervals (CI) for the association between type 2 diabetes (T2D) and various cardiovascular diseases (CVDs).

**Figure 3 biomolecules-14-01467-f003:**
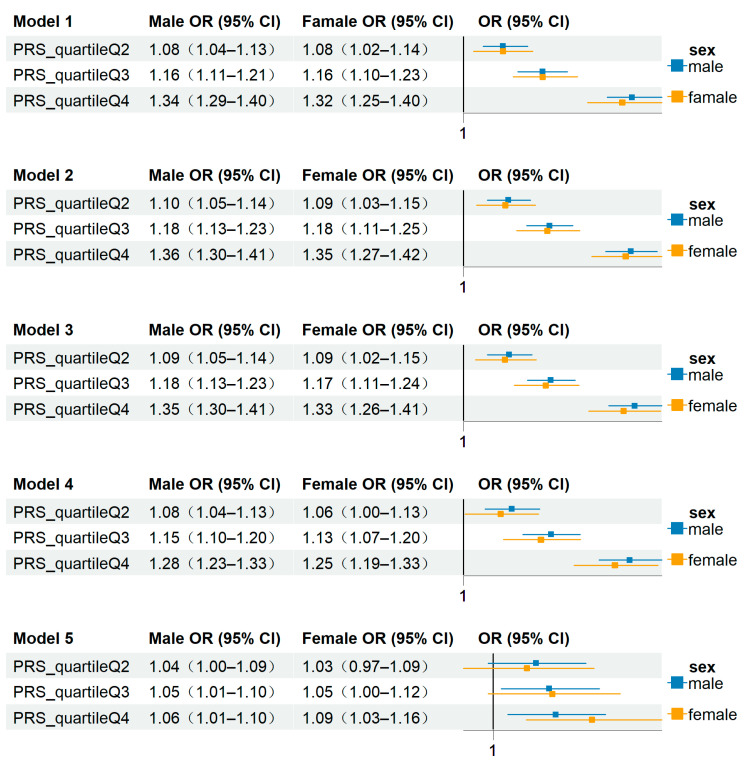
Odds ratios (OR) and 95% confidence intervals (CI) for the association between polygenic risk score (PRS) quartiles (compared to the lowest genetic risk quartile) and cardiovascular disease (CVD) risk by sex across five models.

**Table 1 biomolecules-14-01467-t001:** Summary of diseases, data sources, number of SNPs analyzed, and case/control numbers.

Disease	Download Website	SNP	Case/Control
Angina	GCST90043955	11,842,647	5786/450,562
Atrial fibrillation	GCST90043977	11,842,647	8404/447,944
Heart failure	GCST90043986	11,842,647	1029/455,319
Myocardial infarction	GCST90043954	11,842,647	8528/447,820
Peripheral vascular disease	GCST90044015	11,842,647	847/455,501
Stroke	GCST90044350	11,842,647	6986/448,317
T2D	https://diagram-consortium.org, accessed on 10 July 2024	23,425,876	74,124/824,006

**Table 2 biomolecules-14-01467-t002:** Summary statistics of demographics and selected clinical measures.

	T2D Population		CVD Population	
	Cases *N* = 12,977	Controls *N* = 352,031	Cases *N* = 32,963	Controls *N* = 332,045
**Demographic characteristics**				
Age				
Mean (SD)	58.87 (7.23)	56.66 (8.03)	60.43 (6.79)	56.37 (8.04)
Sex n (%)				
Males	8232 (63.44%)	160,378 (43.94%)	22,135 (67.15%)	146,475 (44.11%)
Females	4745 (36.56%)	191,653 (54.44%)	10,828 (32.5%)	185,570 (55.89%)
Level of education n (%)				
7 yrs	3241 (24.97%)	58,491 (16.62%)	9053 (27.46%)	52,679 (15.87%)
10 yrs	3498 (26.96%)	96,046 (27.28%)	7857 (23.84%)	91,687 (27.61%)
13 yrs	1471 (11.34%)	43,465 (12.35%)	3521 (10.68%)	41,415 (12.47%)
15 yrs	760 (5.86%)	17,809 (5.06%)	1884 (5.72%)	16,685 (5.02%)
19 yrs	1173 (9.04%)	22,852 (6.49%)	2778 (8.43%)	21,247 (6.40%)
20 yrs	2834 (21.84%)	113,368 (32.20%)	7870 (23.88%)	108,332 (32.63%)
**Hospital admission examinations**				
Body mass index (BMI)				
Mean (SD)	31.31 (5.43)	27.26 (4.67)	28.62 (4.86)	27.28 (4.73)
Systolic blood pressure (SBP)	144.47 (18.24)	139.70 (19.02)	141.65 (19.03)	139.69 (19.00)
Diastolic blood pressure (DBP)	83.95 (10.27)	82.17 (10.31)	81.33 (10.59)	82.32 (10.29)
**Lifestyle factors**				
Smoking n (%)				
Yes	1588 (12.24%)	35,646 (10.13%)	4175 (12.67%)	33,059 (9.96%)
No	11,389 (87.76%)	316,385 (89.87%)	28,788 (87.33%)	298,986 (90.04%)
Drinking n (%)				
Yes	12,265 (94.51%)	339,748 (96.51%)	31,597 (95.86%)	320,416 (96.50%)
No	712 (5.49%)	12,283 (3.49%)	1366 (4.14%)	11,629 (3.50%)
**Biological variables**				
Red blood cell count (RBC)	4.65 (0.40)	4.51 (0.40)	4.57 (0.41)	4.51 (0.40)
White blood cell count (WBC)	7.54 (1.85)	6.85 (2.00)	7.24 (1.97)	6.84 (2.00)
Platelet count (PLT)	247.33 (59.78)	253.26 (58.95)	243.34 (59.73)	254.01 (58.83)
Creatinine	74.24 (18.76)	72.28 (17.58)	78.16 (22.48)	71.77 (16.97)
Cholesterol	5.11 (1.23)	5.72 (1.11)	5.13 (1.22)	5.76 (1.09)
High-density lipoprotein (HDL) cholesterol	1.23 (0.29)	1.45 (0.36)	1.31 (0.33)	1.46 (0.36)
Low-density lipoprotein (LDL) cholesterol	3.17 (0.92)	3.58 (0.84)	3.17 (0.92)	3.60 (0.83)
Triglycerides (TG)	2.27 (1.27)	1.72 (0.98)	1.89 (1.07)	1.72 (0.99)
C-reactive protein (CRP)	3.40 (4.65)	2.50 (4.26)	2.88 (4.90)	2.49 (4.21)
Glycated haemoglobin (HbA1c)	46.30 (12.37)	35.63 (5.84)	38.34 (9.05)	35.78 (6.14)
Apolipoprotein A (Apo-A)	0.97 (0.25)	1.04 (0.23)	0.95 (0.24)	1.04 (0.23)
Apolipoprotein B (Apo-B)	1.52 (0.28)	1.54 (0.27)	1.51 (0.28)	1.54 (0.27)
Vitamin D	44.22 (19.10)	49.29 (20.11)	5.13 (1.22)	5.76 (1.09)

**Table 3 biomolecules-14-01467-t003:** Multivariable logistic regression analysis of CVD risk across different PRS quartiles.

	PRS Quartile	OR	95% CI		*p*-Value
			Lower	Upper	
Model 1	PRS_quartileQ2	1.09	1.04	1.12	<0.001
	PRS_quartileQ3	1.16	1.12	1.19	<0.001
	PRS_quartileQ4	1.33	1.29	1.38	<0.001
Model 2	PRS_quartileQ2	1.09	1.06	1.13	<0.001
	PRS_quartileQ3	1.18	1.14	1.22	<0.001
	PRS_quartileQ4	1.35	1.31	1.40	<0.001
Model 3	PRS_quartileQ2	1.09	1.05	1.13	<0.001
	PRS_quartileQ3	1.18	1.14	1.22	<0.001
	PRS_quartileQ4	1.35	1.30	1.39	<0.001
Model 4	PRS_quartileQ2	1.07	1.04	1.11	<0.001
	PRS_quartileQ3	1.14	1.10	1.18	<0.001
	PRS_quartileQ4	1.27	1.23	1.31	<0.001
Model 5	PRS_quartileQ2	1.04	1.00	1.07	<0.05
	PRS_quartileQ3	1.05	1.02	1.09	<0.05
	PRS_quartileQ4	1.07	1.03	1.11	<0.001

## Data Availability

Summary data for genetic associations with type 2 diabetes have been contributed by the DIAGRAM consortium (https://diagram-consortium.org/downloads.html; accessed 10 July 2024). GWAS-Summary data from Jiang, L et al., 2021 [31], and data on angina, heart failure (HF), myocardial infarction (MI), peripheral vascular disease (PVD), and stroke were accessed via the GWAS catalog. The data that support the findings of this study are available from the UK Biobank upon approved request. This research was conducted using resources from the UK Biobank under approved application number 95259.

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
