# Peer review of "Exploring the Genetic Relationship Between Type 2 Diabetes and Cardiovascular Disease: A Large-Scale Genetic Association and Polygenic Risk Score Study"

_biomolecules, 2024, doi:10.3390/biom14111467_

Round 1
Reviewer 1 Report
Comments and Suggestions for Authors
Both diabetes and cardiovascular diseases are the leading cause of death in the world, especially in highly developed countries. Therefore, the subject of the work undertaken by the authors is extremely important from the point of view of public health.
The authors have demonstrated a genetic link between type 2 diabetes and cardiovascular diseases in a large group of people from the UK population.
Congratulations on such extensive research, the results of which may contribute to a better understanding of the link between the diseases described in the manuscript and possible attempts to eliminate the mechanisms that increase the risk of developing type 2 diabetes and cardiovascular diseases.
However, I have a question:
Can the results of the research be applied to the entire white population, or only to people living in the UK? Is there data on differences in the white population living in different areas of Eastern and Western Europe? If so, I think it is worth including such information in the manuscrypt.
Author Response
Dear Reviewer:
We are very grateful to your comments for our manuscript entitled “Exploring the genetic relationship between type 2 diabetes and cardiovascular disease: a large-scale genetic association and polygenic risk score study” (Manuscript ID: biomolecules-3261129). Those comments are all valuable and very helpful in revising and improving our paper. We have carefully read your comments and revised them.
Responds to the reviewer’s comments:
Both diabetes and cardiovascular diseases are the leading cause of death in the world, especially in highly developed countries. Therefore, the subject of the work undertaken by the authors is extremely important from the point of view of public health.
The authors have demonstrated a genetic link between type 2 diabetes and cardiovascular diseases in a large group of people from the UK population.
Congratulations on such extensive research, the results of which may contribute to a better understanding of the link between the diseases described in the manuscript and possible attempts to eliminate the mechanisms that increase the risk of developing type 2 diabetes and cardiovascular diseases.
However, I have a question:
Can the results of the research be applied to the entire white population, or only to people living in the UK? Is there data on differences in the white population living in different areas of Eastern and Western Europe? If so, I think it is worth including such information in the manuscrypt.
Comments 1: Can the results of the research be applied to the entire white population, or only to people living in the UK? Is there data on differences in the white population living in different areas of Eastern and Western Europe? If so, I think it is worth including such information in the manuscrypt.
Response 1: We highly appreciate the reviewer’s valuable comments. We very much agree with you that research on genetic associations between T2D and CVD is of public health importance. In response to your question about whether it can be generalized to the white population as a whole, we very much understand the concerns about the applicability of the study results. Regarding data on whether there are differences in the white population across regions (e.g., Eastern and Western Europe), we found some information in the existing literature that there are indeed some regional genetic and environmental differences that may affect disease associations. However, the data for our study came mainly from a large body of biological and medical data from people living in the UK, so we do have some limitations in extrapolating our conclusions, and are not yet able to fully ascertain the impact of these differences on our findings. Following your suggestions, we have added relevant information about such differences to the manuscript and discuss the need for further validation of the findings in other regions in the future. We have re-written this part according to the Reviewers suggestion. [Page 12, Line 409-410, Some studies have identified genetic differences between populations in different regions, which may influence the associations between diseases [1,2]. Future research should aim to replicate this study in ethnically and geographically diverse populations to enhance the validity and applicability of the results globally.].
We appreciate for your warm work earnestly, and hope that the correction will meet with approval. Once again, thank you very much for your comments and suggestions.
References
- Moskvina, V.; Smith, M.; Ivanov, D.; Blackwood, D.; StClair, D.; Hultman, C.; Toncheva, D.; Gill, M.; Corvin, A.; O’Dushlaine, C. Genetic differences between five European populations. Human heredity 2010, 70, 141-149.
- Vaulin, A.; Karpulevich, E.; Kasianov, A.; Morozova, I. Europeans and Americans of European origin show differences between their biological pathways related to the major histocompatibility complex. Scientific Reports 2024, 14, 21816.

Reviewer 2 Report
Comments and Suggestions for Authors
After reviewing the paper on the genetic relationship between type 2 diabetes (T2D) and cardiovascular disease (CVD), here are some of my constructive comments, suggested revisions, and potential limitations to address:
Strengths and Positive Aspects of the study:
- The study leverages large-scale datasets (UK Biobank and DIAGRAM) to explore associations between T2D and CVDs using a robust sample size.
- The use of both phenotypic association analyses and genetic risk models (LDSC and PRS) enhances the validity of the findings by combining different analytical perspectives.
- The study identifies significant genetic associations between T2D and several CVD subtypes, advancing understanding of the genetic overlap.
Suggested Revisions and Enhancements
- Clarify the Causality Aspect: While the study identifies genetic associations, a more explicit discussion on causality would enhance interpretability. It would be valuable to discuss limitations in establishing causation given the correlational design and suggest any prospective genetic studies that could further clarify causal links.
- Expand on Methodological Rationale: A more detailed explanation of the choice of the LDSC and PRS methods and their respective advantages would strengthen the study. For instance, briefly contrast LDSC with other genetic association methods and discuss why PRS was chosen for risk prediction in this context.
- Control for Potential Confounders: While adjustments are made for lifestyle factors, further detail on how well these adjustments account for other environmental or lifestyle factors specific to CVD and T2D would be beneficial. Future analyses could incorporate a wider array of environmental variables (e.g., diet, physical activity specifics) if available.
- Subgroup Analysis by Ethnicity: The study primarily focuses on European populations. For broader applicability, encourage more detailed stratification by genetic background if data allows, or suggest further studies to verify these findings across ethnic groups to assess generalizability.
- Interpretation of Negative Findings (AF and PVD): Expand on the lack of association between T2D and certain CVD subtypes (e.g., atrial fibrillation and peripheral vascular disease). This could involve speculating why these subtypes may have weaker genetic ties to T2D and proposing future research directions to validate or refute these findings.
- Potential Clinical Application: While the PRS models show predictive potential, the study could provide more discussion on how these scores might be incorporated into clinical practice. This could involve noting limitations in current PRS applications and suggesting research into integrating genetic scores with traditional clinical risk factors.
Identified some Limitations in the study:
- As the study population is mainly European, the findings may not generalize to other populations. Future research should aim to replicate this study in diverse populations to enhance the validity and applicability of the results globally.
- The use of large datasets can sometimes result in inflated associations due to subtle population stratification or residual confounding. To address this, the study could apply more rigorous controls for population structure or perform sensitivity analyses to check the robustness of the associations.
- Self-reported data for certain conditions might introduce bias. Encouraging the use of more robust, clinically verified data in future research could improve data quality.
- Given the complex interplay of genetic and environmental factors in T2D and CVD, the study could improve by discussing gene-environment interactions in the analysis. Suggestions include using statistical models that can accommodate such interactions in future studies.
- Since the PRS models were developed and validated within a single cohort, there is a risk of overfitting to the UK Biobank sample. Future research could enhance model reliability by validating these PRS findings in other datasets or external cohorts.
Author Response
Dear Reviewer:
We are very grateful to your comments for our manuscript entitled “Exploring the genetic relationship between type 2 diabetes and cardiovascular disease: a large-scale genetic association and polygenic risk score study” (Manuscript ID: biomolecules-3261129). Those comments are all valuable and very helpful in revising and improving our paper.
Responds to the reviewer’s comments:
Suggested Revisions and Enhancements
Comments 1: Clarify the Causality Aspect: While the study identifies genetic associations, a more explicit discussion on causality would enhance interpretability. It would be valuable to discuss limitations in establishing causation given the correlational design and suggest any prospective genetic studies that could further clarify causal links.
Response 1: Thank you for your insightful suggestion regarding the discussion of causation. We acknowledge your concerns and concur that clarifying causal relationships will enhance the interpretability of our results. Given that this study employed a correlational design, we recognize its limitations in directly inferring causal relationships. Consequently, we have added a section to the revised manuscript that further discusses these limitations, explicitly stating that the findings reveal only genetic associations rather than causation, and that prospective genetic studies aimed at elucidating causation would be beneficial. [Page 12, Line 412-415, Finally, this study has notable limitations regarding causation. As a correlational study, this research revealed a genetic association between T2D and CVD, however, it does not establish a direct causal relationship.] Based on your recommendation, we have made the simplest causal inference, which initially indicated a causal relationship (Table 1, Figure 1). Currently, our research focuses primarily on genetic associations and risk prediction. In future research, we intend to explore more comprehensive causal inference methods to verify whether the genetic link between these diseases is indeed causal. Thank you once again for your suggestions.
Table 1 Results of MR analysis of the association between type 2 diabetes mellitus and cardiovascular diseases
|
Exposure |
Outcome |
Analysis Method |
SNP |
OR (95% CI) |
P |
|
T2D |
MI |
Inverse variance weighted |
93 |
1.134(1.047~1.228) |
0.002 |
|
|
MI |
MR Egger |
93 |
1.145(0.954~1.375) |
0.149 |
|
|
MI |
Weighted median |
93 |
1.188(1.095~1.289) |
<0.0011 |
|
|
Angina |
Inverse variance weighted |
93 |
1.068(0.988~1.155) |
0.099 |
|
|
Angina |
MR Egger |
93 |
1.127(0.943~1.348) |
0.193 |
|
|
Angina |
Weighted median |
93 |
1.169(1.067~1.281) |
0.001 |
|
|
AF |
Inverse variance weighted |
93 |
0.953(0.900~1.008) |
0.091 |
|
|
AF |
MR Egger |
93 |
0.989(0.870~1.124) |
0.865 |
|
|
AF |
Weighted median |
93 |
0.970(0.885~1.064) |
0.521 |
|
|
HF |
Inverse variance weighted |
93 |
1.151(0.994~1.333) |
0.061 |
|
|
HF |
MR Egger |
93 |
1.053(0.753~1.473) |
0.761 |
|
|
HF |
Weighted median |
93 |
1.093(0.889~1.344) |
0.400 |
|
|
PVD |
Inverse variance weighted |
93 |
1.342(1.143~1.576) |
<0.001 |
|
|
PVD |
MR Egger |
93 |
1.401(0.971~2.021) |
0.075 |
|
|
PVD |
Weighted median |
93 |
1.128(0.874~1.457) |
0.355 |
|
|
Stroke |
Inverse variance weighted |
93 |
1.065(1.011~1.122) |
0.017 |
|
|
Stroke |
MR Egger |
93 |
1.069(0.950~1.203) |
0.271 |
|
|
Stroke |
Weighted median |
93 |
1.088(0.989~1.198) |
0.084 |

Figure 1 Results of MR analysis of the association between type 2 diabetes mellitus and cardiovascular diseases
Comments 2: Expand on Methodological Rationale: A more detailed explanation of the choice of the LDSC and PRS methods and their respective advantages would strengthen the study. For instance, briefly contrast LDSC with other genetic association methods and discuss why PRS was chosen for risk prediction in this context.
Response 2: Thank you for your suggestions regarding the methodology section. We wholeheartedly agree with the importance of incorporating detailed explanations of the LDSC and PRS method selection, along with their respective advantages, to improve the rationale and clarity of the study. In the revised manuscript, we have expanded the methods section significantly. We have included a brief comparison of LDSC with other genetic association methods [Page 4, Line 168-176, LDSC is particularly appropriate for large-scale studies as it does not require individual-level data, making it more efficient than methods such as Genome-wide Complex Trait Analysis (GCTA) and Local Analysis of [co]Variant Association (LAVA), both of which rely on individual-level data to estimate genetic variation and trait correlations. While GCTA focuses on estimating these genetic parameters, LAVA specifically analyzes the local ancestry of admixed populations to investigate the influence of genetic variation on ancestral origins at specific loci [1,2]. In contrast, LDSC is better suited to elucidate the shared genetic architecture between T2D and CVD based on GWAS summary statistics. ].
Additionally, we have elaborated on the selection of risk prediction models by providing further justification for choosing the PRS and offering an introduction and brief comparison of PRS methods [Page 2, Line 74-77, The C + T method is recognized as one of the simplest and most intuitive techniques for constructing PRS. Two common software programs, PLINK and PRSice, can be employed to implement the C + T method. Recently, Choi et al. developed a new software, PRSice-2, available at https://www.prsice.info [3]. ] [Page 4, Line 191-193, Page 5, Line 194-196, We selected PRS for risk prediction because of its ability to efficiently quantify individual genetic risk in large populations without being limited to individual gene associations. PRS is better suited to provide a comprehensive risk assessment than localized analysis methods, thus providing a more practical predictive tool for understanding complex diseases such as T2D and CVD. ]. We believe that these enhancements will contribute to the clarity and persuasiveness of the methodology section. Thank you for your valuable feedback!
Comments 3: Control for Potential Confounders: While adjustments are made for lifestyle factors, further detail on how well these adjustments account for other environmental or lifestyle factors specific to CVD and T2D would be beneficial. Future analyses could incorporate a wider array of environmental variables (e.g., diet, physical activity specifics) if available.
Response 3: Thank you for your valuable suggestions on controlling for potential confounders. We have adjusted for major lifestyle factors (e.g., smoking and alcohol consumption) in our study, but we recognize that these adjustments do not yet cover other environmental or lifestyle factors specific to CVD and T2D. We agree that the inclusion of a wider range of environmental variables (e.g., specific information on diet, physical activity, etc.) would help to further enhance the accuracy and interpretability of the analysis. In future studies, we will endeavor to collect more detailed data in order to more fully control for potential confounders and to explore the relationship between these environmental factors and disease risk in greater depth. [Page 11, Line 403-407, Second, although this study adjusted for lifestyle factors, mainly including smoking status and alcohol consumption, there are other potential environmental or lifestyle factors (e.g., diet, physical activity specifics) that may play an important role in the association between CVD and T2D. Future studies should collect these additional lifestyle data whenever possible and incorporate them into their analyses.] Thank you for your constructive suggestions, which will help improve the rigor of the study and the applicability of the results.
Comments 4: Subgroup Analysis by Ethnicity: The study primarily focuses on European populations. For broader applicability, encourage more detailed stratification by genetic background if data allows, or suggest further studies to verify these findings across ethnic groups to assess generalizability.
Response 4: Thank you for your interest in the applicability of the study. We recognize the limitations of this study, which was based primarily on a European population, and does have some limitations in terms of global applicability. To improve the generalizability of the results, we further emphasize this point in the Discussion section and suggest that future studies could include more detailed stratified analyses by different genetic backgrounds, as data allow. In addition, we also suggest that similar studies be conducted in various groups residing in different geographies to verify the applicability and robustness of the results of this study. [Page 11, Line 407, Page 12, Line 408-412, Furthermore, as the study population is mainly European, the findings may not generalize to other populations. Some studies have identified genetic differences between populations in different regions, which may influence the associations between diseases [4,5]. Future research should aim to replicate this study in ethnically and geographically diverse populations to enhance the validity and applicability of the results globally.] Thank you for your constructive comments, which will help to enhance the value of the study for a wide range of applications in diverse populations.
Comments 5: Interpretation of Negative Findings (AF and PVD): Expand on the lack of association between T2D and certain CVD subtypes (e.g., atrial fibrillation and peripheral vascular disease). This could involve speculating why these subtypes may have weaker genetic ties to T2D and proposing future research directions to validate or refute these findings.
Response 5: Thank you for your insight into our study regarding the lack of association between T2D and certain CVD subtypes such as AF and PVD. In response to your suggestion, we have provided further discussion of these negative results in the revised manuscript. [Page 10, Line 326-342, Some studies indicate that the genetic association between T2D and AF may not be significant, possibly due to the mediating role of systolic blood pressure in this relationship [6]. Although T2D patients have a significantly increased risk of both macrovascular and microvascular complications in the development of AF, there is no direct evidence linking T2D to an increased prevalence of AF [7]. Nevertheless, some studies have shown that T2D may significantly increase the risk of mortality, coronary heart disease and HF in patients with AF. This risk is particularly pronounced in elderly patients with T2D, where AF significantly raises the absolute risk of cerebrovascular events [8]. Thus, while T2D may not directly induce AF, effective blood glucose management remains crucial for AF patients to mitigate the risk of complications. Similarly, existing evidence does not support the notion that T2D directly causes PVD. However, research indicates that patients both T2D and PVD experience a significantly greater risk of all-cause mortality and cardiovascular death compared to those with CVD alone [9]. Thus, although T2D does not seem to play a significant direct role in the onset of PVD, clinical management should still prioritize monitoring blood glucose levels in T2D patients to reduce the risk of comorbidities.]
We point out that existing studies suggest that the genetic association between type 2 diabetes and AF may be weak, which may be related in part to the mediating role of systolic blood pressure. Furthermore, although type 2 diabetes is associated with a significantly increased risk of complications in patients with AF, there is no evidence that it directly increases the prevalence of AF. For PVD, although T2D does not directly contribute to its development, patients with T2D combined with PVD have a significantly increased risk of death. We believe that future studies could further explore the mechanisms and applicability of these results. We thank you for your valuable suggestions and hope that the revised interpretation will lead to clearer and more rigorous conclusions of the study.
Comments 6: Potential Clinical Application: While the PRS models show predictive potential, the study could provide more discussion on how these scores might be incorporated into clinical practice. This could involve noting limitations in current PRS applications and suggesting research into integrating genetic scores with traditional clinical risk factors.
Response 6: Thank you for your valuable comments. In the revised manuscript, we have further discussed how to integrate PRS with traditional clinical risk factors. Although our study demonstrates the potential of PRS in predicting CVD risk, we also recognize the limitations of the current PRS models, especially since they were developed and validated in the UK Biobank, and there may be a risk of overfitting. To enhance the reliability of the PRS models, future studies could further validate these findings by performing validation in other datasets or external cohorts. In addition, we suggest that future studies could explore ways to combine genetic scores with traditional clinical risk factors to more comprehensively assess an individual's disease risk, thereby advancing the practical application of these scores in clinical practice. [Page 12, Line 429-433, Although PRS models have demonstrated enhanced potential in predicting CVD risk, its clinical application warrants further investigation. Future research should focus on developing targeted strategies for the effective integration of PRS scores into clinical practice, as well as exploring methods to combine these scores with traditional clinical risk factors to achieve a more comprehensive risk assessment. ] Thank you again for your suggestions, and we will continue to improve this section.
Identified some Limitations in the study:
Comments 1: As the study population is mainly European, the findings may not generalize to other populations. Future research should aim to replicate this study in diverse populations to enhance the validity and applicability of the results globally.
Response 1: Thank you for your suggestions. We fully understand your concerns about sample limitations. As the sample in this study was primarily from a European population, it may indeed affect the general applicability of the results. In future studies, we will prioritize data collection and analysis in more ethnic and geographic contexts to increase the global applicability and credibility of the findings. [Page 12, Line 407-409]
Comments 2: The use of large datasets can sometimes result in inflated associations due to subtle population stratification or residual confounding. To address this, the study could apply more rigorous controls for population structure or perform sensitivity analyses to check the robustness of the associations.
Response 2: Thank you for your suggestions. We recognize that the use of large datasets may introduce subtle population stratification or residual confounders. In this study, we utilized baseline data from participants' initial assessments conducted between 2006 and 2010 within the UKB. The study population was limited to individuals who satisfied the UKB's internal genetic quality control criteria (UKB Field 22020) and self-identified as White British (UKB Field 21000). In future studies, we plan to apply even more refined genetic quality control measures and include additional ancestry groups to assess the robustness of our results across a more diverse population base. This approach would not only enhance the generalizability of our findings but also further mitigate any potential bias arising from population stratification in large datasets.
Comments 3: Self-reported data for certain conditions might introduce bias. Encouraging the use of more robust, clinically verified data in future research could improve data quality.
Response 3: Thank you for your suggestions. Self-reported data can indeed lead to bias, especially in the accuracy of lifestyle-related data. In future studies, we will use clinically validated, objective data sources, such as medical records or biological samples, whenever possible to improve the accuracy of the data and the credibility of the findings.
[Page 11, Line 397-403, First, some of the self-reported data utilized may be biased, particularly in relation to lifestyle factors such as smoking and alcohol consumption. Self-reported data are affected by factors such as respondents' memory bias and social desirability bias, and may not accurately reflect actual behavior. To enhance data quality, future studies should prioritize the use of more reliable and clinically validated data sources, such as medical records or biospecimens, to minimize potential biases and improve the credibility of the findings. ]
Comments 4: Given the complex interplay of genetic and environmental factors in T2D and CVD, the study could improve by discussing gene-environment interactions in the analysis. Suggestions include using statistical models that can accommodate such interactions in future studies.
Response 4: Thank you for your suggestions. The discussion you presented on gene-environment factors is very valuable. Future studies will attempt to incorporate gene-environment factors models to more fully explore the complex interplay of genetic and environmental factors in T2D and CVD, which can contribute to more accurate risk prediction and individualized intervention strategies. [Page 12, Line 433-438, Given the complex interplay between genetic and environmental factors in T2D and CVD, future research could further improve the understanding of these relationships through statistical modeling of gene-environment interactions. Such models can help explore the moderating effects of different environmental factors on PRS and provide more basis for personalized risk assessment and intervention strategies. ]
Comments 5: Since the PRS models were developed and validated within a single cohort, there is a risk of overfitting to the UK Biobank sample. Future research could enhance model reliability by validating these PRS findings in other datasets or external cohorts.
Response 5: Thank you for your reminder. We recognize the potential risk of overfitting in developing and validating PRS models in a single cohort. Future studies will prioritize validating these PRS findings in different datasets or external cohorts to enhance the generalization of the model and ensure its broader applicability. [Page 12, Line 438-444, Furthermore, as PRS models are developed and validated based on a single cohort, there is a risk of overfitting, particularly concerning the UK Biobank sample. To enhance the reliability of the PRS model, it is imperative that future studies validate these findings across different datasets or external cohorts. This will help to validate the ability of the PRS model to generalize across different populations and provide stronger evidence to support its wider application. ]
We appreciate for your warm work earnestly, and hope that the correction will meet with approval. Once again, thank you very much for your comments and suggestions.
References
- Werme, J.; van der Sluis, S.; Posthuma, D.; de Leeuw, C.A. An integrated framework for local genetic correlation analysis. Nature genetics 2022, 54, 274-282.
- Yang, J.; Lee, S.H.; Goddard, M.E.; Visscher, P.M. GCTA: a tool for genome-wide complex trait analysis. The American Journal of Human Genetics 2011, 88, 76-82.
- Choi, S.W.; O'Reilly, P.F. PRSice-2: Polygenic Risk Score software for biobank-scale data. Gigascience 2019, 8, giz082.
- Moskvina, V.; Smith, M.; Ivanov, D.; Blackwood, D.; StClair, D.; Hultman, C.; Toncheva, D.; Gill, M.; Corvin, A.; O’Dushlaine, C. Genetic differences between five European populations. Human heredity 2010, 70, 141-149.
- Vaulin, A.; Karpulevich, E.; Kasianov, A.; Morozova, I. Europeans and Americans of European origin show differences between their biological pathways related to the major histocompatibility complex. Scientific Reports 2024, 14, 21816.
- Reddy, R.K.; Ardissino, M.; Ng, F.S. Type 2 Diabetes and Atrial Fibrillation: Evaluating Causal and Pleiotropic Pathways Using Mendelian Randomization. Journal of the American Heart Association 2023, 12, e030298.
- Geng, T.; Wang, Y.; Lu, Q.; Zhang, Y.-B.; Chen, J.-X.; Zhou, Y.-F.; Wan, Z.; Guo, K.; Yang, K.; Liu, L. Associations of new-onset atrial fibrillation with risks of cardiovascular disease, chronic kidney disease, and mortality among patients with type 2 diabetes. Diabetes Care 2022, 45, 2422-2429.
- Matsumoto, C.; Ogawa, H.; Saito, Y.; Okada, S.; Soejima, H.; Sakuma, M.; Masuda, I.; Nakayama, M.; Doi, N.; Jinnouchi, H. Incidence of atrial fibrillation in elderly patients with type 2 diabetes mellitus. BMJ Open Diabetes Research and Care 2022, 10, e002745.
- Avdic, T.; Carlsen, H.K.; Rawshani, A.; Gudbjörnsdottir, S.; Mandalenakis, Z.; Eliasson, B. Risk factors for and risk of all-cause and atherosclerotic cardiovascular disease mortality in people with type 2 diabetes and peripheral artery disease: an observational, register-based cohort study. Cardiovascular Diabetology 2024, 23, 127.
